# Evaluation of Color, Phytochemical Compounds and Antioxidant Activities of Mulberry Fruit (*Morus alba* L.) during Ripening

**Surapon Saensouk** [1,2]**, Rattanavalee Senavongse** [2,3]**, Chanakran Papayrata** [2,4] **and Theeraphan Chumroenphat** [2,4,]***

1   Walai Rukhavej Botanical Research Institute, Mahasarakham University, Kantarawichai District, Maha Sarakham 44150, Thailand
2   Diversity of Family Zingibeaceae and Vascular Plant of Its Applications Research Unit, Mahasarakham University, Kantarawichai District, Maha Sarakham 44150, Thailand
3   School of Crop Production Technology, Institute of Agricultural Technology, Suranaree University of Technology, Nakhon Ratchasima 30000, Thailand
4   Laboratory Equipment Center, Division of Research Facilitation and Dissemination, Mahasarakham University, Kantarawichai District, Maha Sarakham 44150, Thailand
*   Correspondence: theeraphan.chum@hotmail.com

**Abstract:** Mulberry fruits are used for food, cosmetics and medicine. Several phytochemical and bioactive compounds in mulberry fruits are widely used for health benefits. During the ripening stage of mulberry fruits, different phytochemicals are present. This study investigates color, phytochemical compounds and antioxidant activity in mulberry fruit during seven ripening stages. The results indicate that the color changes from green to purple. The green stage contains high levels of vitamin c (12 mg/gDW), while the purple stage has the highest sugar levels, particularly fructose (241 mg/gDW) and glucose (171 mg/gDW). Trends of amino acids, anthocyanin and vitamin C increased initially, peaking in M6 and then declined in the final ripening stage (M7). Phenolic acid, flavonoids and γ-aminobutyric increased when the ripening level increased. The antioxidants activity was analyzed by 2,2-diphenyl-1-picryl-hydrazyl-hydrate (DPPH) free radical and ferric reducing antioxidant power (FRAP) assays. It was found to have the highest contents in M7 at 19 mgTE/gDW (DPPH) and 22 mg $FeSO_4$/gDW (FRAP), which were positively correlated with the total phenolic and total flavonoid contents. This study provides information on mulberry fruit during ripening that should be helpful in designing products while maintaining its high antioxidant content, excellent bioactivity and quality for use in food, cosmetics and medicine.

**Keywords:** phenolic acids; flavonoids; antioxidant; sugar; ripening stage

## 1. Introduction

The mulberry (*Morus alba* L.) belongs to the Moraceae family and genus *Morus* and it is wildly grown from the tropics to the subarctic, such as China, India and Thailand [1–3]. There are many uses for all parts of the mulberry plant (leaf, stem, root and fruit), which are used in food, cosmetics and folk medicine [4–6]. Specifically, mulberry fruits have been used for a long time as fresh fruit and in food products (jam, juice, wine, tea and mulberry fruit powder, etc.) [7]. Additionally, mulberry fruit is used as a medicinal plant with various activities as previously reported. Several phytochemical compounds of mulberry fruit have been reported by many researchers [1–5]. Mulberry fruit contains different chemical components at different ripening stages, which leads to a wide use of many mulberry fruits, especially due to its health benefits involving antioxidants, anti-inflammatory, anti-diabetic and antimicrobial properties. However, the chemical components differ in this fruit deepening on geographic location, the season it is harvested, plant species and maturity [8]. Mulberry fruit in Thailand may have different chemical compounds with antioxidant

activity than mulberry fruits in other countries [3]. Each ripening stage of mulberry fruit has different major compounds due to the synthesis pathway to maturity in the fruit [9–11]. As such, each stage of ripening results in different benefits useful in different applications.

There have been several reports of mulberry fruit with different colors (green, red, purple, white and black colors) [12]. Several phytochemical and bioactive compounds, such as vitamin C, phenolic acid (chlorogenic acid, caffeic acid) [13], flavonoid compounds (rutin, quercetin and kaempferol) [8] and anthocyanin abundant in cyanindin-3-O-glucoside [7,14], have been reported in mulberry fruit. These bioactive compounds are correlated positively with biological activities due to their antioxidants and anti-inflammatory and anti-diabetic properties [15]. There have been several reports on other chemical components, like amino acids, minerals and fats [13]. While in Thailand, mulberry fruit is widely used, there are few studies revealing the ripening stage that can provide useful information for various applications.

In this study, we investigated the major compounds in the seven stages of mulberry fruit ripening, including color, vitamin C, sugar, phenolic acids, flavonoids, amino acids (The importance of the studies in the mulberry fruits was presented in Table S2) and antioxidant activity. This study will primarily offer relevant information for the expanded usage of mulberry fruit in each of its seven stages of ripening. Mulberry fruit can be used as a functional food ingredient, cosmetic active compound and pharmaceutical active compound with biological activity.

## 2. Materials and Methods

### 2.1. Plant Materials and Sample Preparation

The part of the mulberry fruit used in this study was the Chaingmai-60 cultivar, grown on a farm by the Silk Innovation Center at Mahasarakham University, Thailand. The fruit was hand-harvested at seven stages of mulberry fruit ripening, which were identified based on fruit color and previous studies (Figure 1) [13]. The temperatures (include minimum and maximum) and relative humidity of the mulberry at fruits each stage were collected from 17 December 2021 to 20 January 2022, as shown in Table 1. The specimens were deposited in the herbarium with the voucher specimen number of CT190201-08. For each ripening stage, 500 g of fresh fruit were measured for skin color. Then, they were washed and dried using a freeze dryer (Scanvac CoolSafe model 100-9 Pro, LaboGene ApS, Lynge, Denmark) until the moisture content of the samples was less than 7%. The ground samples were sieved through a 40-mesh wire sieve and kept at −20 °C until they were analyzed for phytochemical compounds and antioxidant activities.

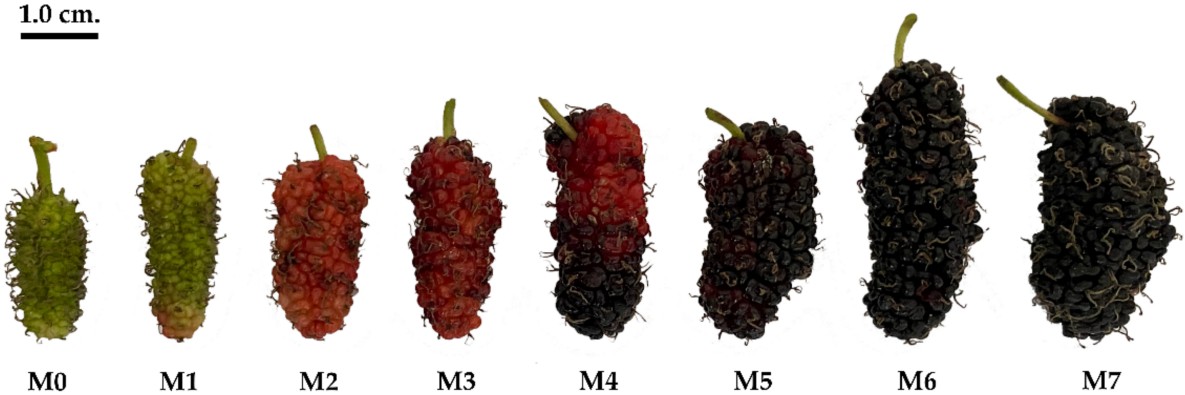

**Figure 1.** Mulberry fruit (*Morus alba* L.) during ripening.

**Table 1.** The minimum to maximum temperatures and relative humidity of mulberry fruits with seven ripening stages.

| Date of Collection | Ripening Stage | Temperature Minimum ($^\circ$C) | Temperature Maximum ($^\circ$C) | Relative Humidity (%) |
|---|---|---|---|---|
| 17 December 2021 | M0 | 28 | 29 | 58 |
| 24 December 2021 | M1 | 27 | 29 | 66 |
| 31 December 2021 | M2 | 25 | 26 | 57 |
| 7 January 2022 | M3 | 27 | 28 | 54 |
| 14 January 2022 | M4 | 25 | 25 | 61 |
| 16 January 2022 | M5 | 26 | 27 | 58 |
| 18 January 2022 | M6 | 25 | 25 | 54 |
| 20 January 2022 | M7 | 24 | 25 | 65 |

M0–M7 are mulberry fruits during different ripening stages.

### 2.2. Color Determination

The change in the color of fresh mulberry fruit during the seven stages of ripening was evaluated with a color instrument (Minolta CR-300 Chroma Meter, Konica Minolta, Osaka, Japan) with *L\**, *a\** and *b\** color scales. Brightness was determined by L-values exhibiting positive and negative numbers; values determined the redness and greenness with positive and negative numbers and yellowness and blueness were shown with b values [16]. A white standard was used to calibrate the instrument. Ten samples were measured individually for each treatment and the average of the 10 measurements was calculated. The color difference $\Delta E$ was calculated from the *L\**, *a\** and *b\** parameters according to Siriamornpun et al. [16].

### 2.3. Total Phenolic Contents (TPC) and Total Flavonoids Contents (TFC)

The TPC and TFC contents were determined according to a previously published protocol [17] using 20% Folin–Ciocalteu reagent. In each well plate, the extracted samples (20 μL) were mixed with Folin–Ciocalteu reagent (100 μL) and kept at 37 $^\circ$C for 4 min. A 10% $Na_2CO_3$ solution was then added to the mixture at 75 μL and kept at room temperature for 2 h. The mixed sample was measured with a microplate reader (Varioskan Lux Multimode microplate reader, Thermo Fisher Scientific, Waltham, MA, USA) at 725 nm using gallic acid as a standard. The total phenolic contents were reported as mg gallic acid equivalents per gram of the dried weight of the sample (mg GAE/gDW). The TFC technique was used on a 96-well plate, with 25 μL of extracted sample added to 100 μL of purified water and 10 μL of 5% $NaNO_2$ solution. A total of 15 μL of a 10% $AlCl_36H_2O$ solution was then added to the mixture after five minutes of shaking, following which, 50 μL of 1 M NaOH and 50 μL of purified water were added. Immediately after the sample was mixed, the absorbance was measured at 510 nm with a Varioskan Lux Multimode microplate reader (Thermo Fisher Scientific, Waltham, MA, USA). The TFC content was represented as milligrams of rutin equivalents per gram of the dried weight of the sample (mg RE/gDW).

### 2.4. Vitamin C by HPLC

The method of extraction and analysis for vitamin C using a HPLC instrument followed that established by Kubolat et al. [18]. Vitamin C was detected at 280 nm using a diode array detector on a Shimadzu LC-20AC series HPLC system (Shimadzu, Tokyo, Japan) with an InertSustain$^{\circledR}$ C18 column (250 mm 4.6 mm i.d., 5 m, GL Sciences Inc., Tokyo, Japan). The concentrations of the vitamin C content were calculated using the corresponding external standards. The result was shown as mg per gram of dried weight of the sample.

### 2.5. Sugar Contents

The extraction and analysis methods for free sugar followed those previously published [19]. The extraction of sugar was performed by mixing 0.1 g of powdered dry samples with 5 mL of distilled water and incubating the mixture in a water bath at 80 $^\circ$C for 30 min. For HPLC analysis, the supernatants from the samples were collected and filtered

through a 0.22 mm filter after being centrifuged at $6300 \times g$ for 10 min. Individual free sugars (sucrose, fructose and glucose) were determined with an HPLC (A Shimadzu 20 series system). The sugars were identified by a Sugar-Pak I column (250 mm × 4.6 mm i.d. 5 mm) with guard column (waters, Milford, USA) at a flow rate of 0.5 mL/min. Deionized water as a mobile phase was eluted using an isocratic condition for 40 min at 80 °C. Detection was carried out with a refractive index detector. The concentrations were calculated using calibration curves constructed previously with the appropriate external standards.

### 2.6. Phenolic Acid and Flavonoid Compounds by HPLC

The phenolics and flavonoids were extracted as previously described [20]. In brief, 1.0 g of each sample was treated with 20 mL of a 1:100 (*v/v*): HCl:methanol solution. The mix was shaken for 12 h at 37 °C. After shaking, the sample was filled with Whatman No. 1 paper filler. The pellets were again extracted using the same method. After being vacuum-dried at 40 °C, the mixed filtrates were redissolved in 5 mL of a 50:50 methanol: water solution. Before HPLC analysis, the samples were filtered through a 0.45-m nylon membrane filter.

The individual phenolic acids and flavonoid compounds were analyzed by HPLC (Shimadzu, Kyoto, Japan), in accordance with the methodology described by Kubola et al. [18]. An InertSustain® C18 (250 mm × 4.6 mm i.d., 5 m: GL Sciences Inc., Tokyo, Japan) column with a guard column was used. A detector with a diode array was used to detect the wavelengths for flavonoids (280, 370 nm) and phenolic acids (280, 320 nm). External standards were used for comparisons with individual phenolic acids and flavonoids in the extracted samples.

### 2.7. Amino Acid Contents by LC/MS/MS

The extraction of the amino acids in the samples were evaluated using a methodology previously described by Chumroenphat et al. [20]. A Shimadzu LCMS-8030 triple-quadrupole mass spectrometer (Kyoto, Japan) operating in ESI mode and a Shimadzu LC-20AC series HPLC system were used to perform the LC/MS/MS (Shimadzu, Kyoto, Japan). The amino acid analysis in isocratic elution was performed on an InertSustain® C18 (2.1 × 150 mm, 3 μm) column connected to a guard column. With a flow rate of 0.2 mL/min and a column temperature of 40 °C, the mobile phase was constituted of solvent A: formic acid 0.1% (*v/v*) in water and solvent B: formic acid 0.1% (*v/v*) in water/methanol (50:50; *v/v*) and the MS/MS was performed in accordance with Chumroenphat and Saensouk [21]. The results achieved were presented as amino acid in dried weight of the sample in μg/g DW ± SD (n = 3). The auto-optimizations are summarized in the Supplementary Materials (Table S1).

### 2.8. γ-Aminobutyric (GABA) and Anthocyanin by LC/MS/MS

The extraction of γ-aminobutyric (GABA) was performed following the method described by Lee and Hwang [13]. In this method, 0.5 g of dried sample was extracted with 10 mL of 3% (*v/v*) trichloroacetic acid and ultrasonicated at 40 °C for 1 h before being centrifuged (Sigma Laborzentrifugen GmbH, Osterode am Harz, Germany) for 5 min at $1717 \times g$. The residue was triplicated for extraction and combined and then adjusted to 25 mL of volume with trichloroacetic acid (3% *v/v*). Filtration with a 0.45 μm nylon filter was performed prior to injection into the HPLC. The anthocyanins were extracted as described by Jorjong et al. [22]. Briefly, 1 g of sample was extracted with 20 mL of methanol at ambient temperature for 20 min using a sonicator bath (KuDos Model 621 OPH, Shanghai, China). The samples were centrifuged at $10,000 \times g$ for 10 min at 4 °C, the solvent was collected and the residue was reextracted with a fresh 10 mL of the solvent. The two extracts from the first and second extractions were combined, and the volume was adjusted to 20 mL with the same solvent. The extracted samples were filtered through a 0.22 μm membrane filter and stored at −18 °C prior to analysis. The anthocyanin compounds were analyzed by LC/MS/MS (Shimadzu LCMS-8030 triple quadrupole mass spectrometer) with the electrospray ionization (ESI) mode together with an HPLC system (Shimadzu, Kyoto, Japan). Anthocyanins were carried to an InertSustain® C18 (2.1 × 150 mm, 3 μm) column coupled with a guard column at a flow rate

of 0.3 mL/min and 35 °C column temperature. The eluents consisted of acetic acid 1% (*v/v*) in deionized water (solvent A) and acetonitrile (solvent B). Anthocyanins were eluted using an isocratic condition. The MS/MS condition was performed as described by Oh et al. [23] for the analysis of anthocyanin. The analysis of GABA with LC/MS/MS was performed according to El-Naggar et al. [24]. The auto-optimizations are shown in the Supplementary Materials (Table S1). The concentrations of anthocyanins and GABA were quantified using external standards.

### 2.9. Antioxidant Activity

- *DPPH free-radical scavenging*

The identification of the scavenging DPPH radicals of the extracts was performed as described by Chumroenphat et al. [15]. In general, 20 μL of the extract or the control was mixed with 180 μL of 60 μM DPPH solution dissolved in methanol. After incubation for 30 min in the dark at ambient temperature, the absorption was measured at 517 nm using a microplate reader (Varioskan Lux, Thermo Fisher Scientific, USA) [17]. The results were expressed as mg Trolox equivalents (TE) per gram of dried wight (mg TE/g DW).

- *Ferric reducing/antioxidant power assay (FRAP)*

The FRAP procedures were performed according to a published method [17]. Briefly, 5 μL of each extracted sample was mixed with 180 μL of FRAP reagent in the 96-well plate. The FRAP reagent was freshly prepared by mixing 0.3 M acetate buffer (pH 3.6) and 10 mM TPTZ in 40 mM HCl and 20 mM $FeCl_3 6H_2O$ in a 10:1:1 ratio at 37 °C. The reagent was incubated at 37 °C for 2 h before use. The mixture was shaken for 1 min and incubated at 37 °C for 15 min. The absorbance was measured at 593 nm against a control. The FRAP values were presented as mg $FeSO_4$ per g of dried weight (mg $FeSO_4$/g DW).

### 2.10. Statistical Analysis

The mean of three replicates, plus one standard deviation (SD), was used to represent all results. The data were analyzed using a one-way ANOVA and the least significant difference (LSD) test. Significant differences between samples were indicated at $p < 0.05$. The Pearson's correlation test was used to assess the correlations among the means. The principal component analysis (PCA) and all statistical analyses were performed using SPSS for Windows.

## 3. Results and Discussion

This study identified the phytochemical and antioxidant activity of mulberry fruit at the seven different stages of ripening. The sample used for the study came from a single source, a single area and a single season. The different ripening stages with different colors, phytochemical compounds and biological activities will be discussed in the following sections.

### 3.1. Changes in Color, Total Phenolic Content, Total Flavonoid Content and Vitamin C of Mulberry Fruit during Ripening

The color of the skin of the fruit was identifying during the seven stages of ripening. The color of each stage was measurement with a color meter, as shown in Table 2. The brightness is indicated by the *L\** value when compared to positive and negative values; the redness, greenness and positive and negative values were identified by the *a\** values, while the yellowness and blueness were determined by the *b\** values. The M0-M1 had the highest *L\** value (brightness), while the negative value of the *a\** value showed a green color. On the other hand, in the next step of the ripening middle stage, the red color increased, as shown in M2–M3. It shows a positive *a\** value from −6.3 to 13.3, while the *L\** and *b\** values decreased. The last of the ripening stages were M4-M7, which showed violet and black colors. The color meter showed a decrease in the *L\**, *a\** and *b\** values, as shown by the decreased bright red and yellow colors. In contrast, the M4-M7 stages showed positive *L\**, *a\** and *b\** values, as shown by the purple to dark purple color, which supports

a previous study reporting that mulberry fruit that had a purple color were sources of anthocyanin [7]. The total color difference ($\Delta E$) is a combination of parameters $L^*$, $a^*$ and $b^*$ values. This result indicates that the extent of color change during ripening of M4 and M5 was not significantly different ($p < 0.05$) with seven stages of fruit ripening. However, other stages of mulberry ripening were different compared with the initial stage (M0). This result can be explained by the ripening of the mulberry fruit as the color changes from green to purple, which is often connected with the formation of anthocyanins, suggesting a modification of pigment density in the surface tissues and the breakdown of chlorophylls and carotenoids [5,25–27].

**Table 2.** Changes in color, total phenolic contents, total flavonoid contents and vitamin C of mulberry fruits during ripening.

| Sample Name | Color | | | | TPC | TFC | Vitamin C |
|---|---|---|---|---|---|---|---|
| | $L^*$ | $a^*$ | $b^*$ | $\Delta E$ | (mgGAE/gDW) | (mgRE/gDW) | (mg/gDW) |
| M0 | $53.62 \pm 0.07$ [a] | $-6.38 \pm 0.07$ [f] | $17.53 \pm 0.17$ [a] | - | $23.21 \pm 0.61$ [g] | $2.95 \pm 0.07$ [f] | $1.86 \pm 0.03$ [h] |
| M1 | $53.87 \pm 0.34$ [a] | $-1.95 \pm 0.15$ [e] | $15.54 \pm 0.06$ [b] | $2.74 \pm 0.14$ [f] | $25.88 \pm 0.28$ [f] | $3.13 \pm 0.02$ [e] | $8.84 \pm 0.91$ [b] |
| M2 | $50.69 \pm 0.20$ [b] | $5.79 \pm 0.28$ [d] | $11.47 \pm 0.26$ [c] | $17.51 \pm 0.88$ [e] | $27.10 \pm 0.53$ [e] | $3.19 \pm 0.14$ [d] | $12.30 \pm 0.51$ [a] |
| M3 | $46.73 \pm 0.05$ [c] | $13.30 \pm 0.63$ [a] | $7.88 \pm 0.10$ [d] | $28.42 \pm 1.42$ [b] | $27.32 \pm 0.46$ [e] | $3.54 \pm 0.06$ [d] | $3.22 \pm 0.09$ [c] |
| M4 | $41.26 \pm 0.25$ [d] | $13.02 \pm 0.16$ [a] | $5.46 \pm 0.07$ [e] | $29.81 \pm 1.49$ [a] | $29.07 \pm 0.87$ [d] | $4.13 \pm 0.35$ [c] | $2.85 \pm 0.02$ [d] |
| M5 | $36.88 \pm 0.45$ [e] | $10.63 \pm 0.25$ [b] | $3.11 \pm 0.46$ [f] | $29.41 \pm 1.47$ [a] | $37.22 \pm 0.44$ [c] | $4.21 \pm 0.15$ [b] | $2.57 \pm 0.02$ [e] |
| M6 | $32.98 \pm 0.03$ [f] | $6.20 \pm 0.14$ [c] | $0.52 \pm 0.05$ [g] | $27.26 \pm 1.36$ [c] | $55.94 \pm 1.69$ [b] | $4.27 \pm 0.13$ [b] | $2.44 \pm 0.01$ [f] |
| M7 | $32.25 \pm 0.21$ [f] | $5.17 \pm 0.34$ [d] | $0.04 \pm 0.00$ [h] | $26.93 \pm 1.35$ [d] | $64.76 \pm 2.16$ [a] | $4.75 \pm 0.22$ [a] | $2.37 \pm 0.02$ [g] |

Values are expressed as mean $\pm$ SD of triplicate measurements (n = 3). M0–M7 are mulberry fruits during different ripening stages; means with different letters are significantly different at $p < 0.05$ within the same column. $L^*$: brightness; $a^*$: redness, greenness; $b^*$: yellowness and blueness.

The total phenolic content (TPC) of mulberry ripening over the seven stages is shown in Table 2. The TPC was found to be in a range of 23.2–64.7 mgGAE/gDW, with M7 having the highest TPC, followed by M6, M5, M4 and M0 having the lowest. These results could be explained by M7 having high phenolic acid from the formation of anthocyanins in the last ripening stage. In the first ripening stage (M0–M2), anthocyanins were absent; instead, there was a higher vitamin C level. This was similar to a previous study reporting that TPC increased from unripened to fully-ripened stages [12,28]. For the TFC of mulberry with different ripening stages, it was found to be highest in M7 (4.75 mgRE/gDW), follow by M6 (4.27 mgRE/gDW), M5 (4.21 mgRE/gDW) and M4 (4.13 mgRE/gDW), while M0 (2.95 mgRE/gDW) had the lowest TFC (Table 2). In this regard, the value of M7 was higher than those reported for mulberry fruits for Pakistan mulberry cultivars (*Morus macroura*: 2.49 mg CE/g) [28], Korean mulberry in five cultivars (0.06 to 0.65 mg CE/g) [29] and China mulberry in 13 cultivars of black mulberry fruit (*M. nigra*) (1.21 RE/g to 2.86 mg RE/gDW ) [14]. Conversely, Pakistan mulberry cultivars (*Morus nigra*: 10.2 mg CE/g) [28] and Turkey mulberry (7.0 mg CE/gDW) [30] were reported as other countries of mulberry fruits with higher results than in the present study. In this regard, we can admit that the difference in TPC and TFC in mulberry fruit could be related to the ecological condition during harvest, genotype, growth condition, genetic diversity and maturity stage of the fruit [1,31–33].

The most important vitamins for human nutrition are commonly considered to be found in fruits and vegetables, particularly vitamin C [34]. The vitamin C content of fruits varies according to variety, species and stage of development [35]. The results are shown in Table 2. It was found that the amount of vitamin C was significantly different ($p < 0.05$) in mulberry fruit within the seven stages of ripening, ranging from 1.6 to 12.3 mg/gDW (Table 2). The highest vitamin C values were found in the M2 ripening stage, which were 12.3 mg/gDW, followed by M1 (8.8 mg/gDW) and M3 (3.2 mg/gDW). The previously reported vitamin C content of Chinese mulberry fruit was 2.5 mg/g DW [36], and that of the Egyptian black mulberry was 1.3 mg/gFW [37]. These contents were lower than those found in the present study, at 4.9 and 9.4-fold, respectively. Vitamin C functions

as an antioxidant and a normal daily consumption of 250–500 mg is recommended for antioxidant activity and removal of free radicals [38].

### 3.2. Sugar Contents

Sugar in fruit is a key quality, as it directly impacts taste, preference and acceptance by the consumer. Natural sugar in fruits can be found as fructose, glucose and sucrose, which are normally predominant. These results show that the values of free sugar (glucose, fructose and sucrose) in the seven ripening stages of mulberry fruits were different and significant ($p < 0.05$). The total free sugar (glucose, fructose and sucrose) values are shown in Figure 2 with the following order: M6 > M7 > M5 > M4 > M3 > M2 > M1 > M0. Glucose and fructose increased in line with the stage of ripeness, while sucrose decreased, supporting the findings of a previous study [39]. These results could be explained by the hydrolyzation of sucrose by invertases or synthases into glucose and fructose during the ripening stages [40,41]. In agreement with our findings, several authors have reported on mulberry fruit and other fruits [41,42].

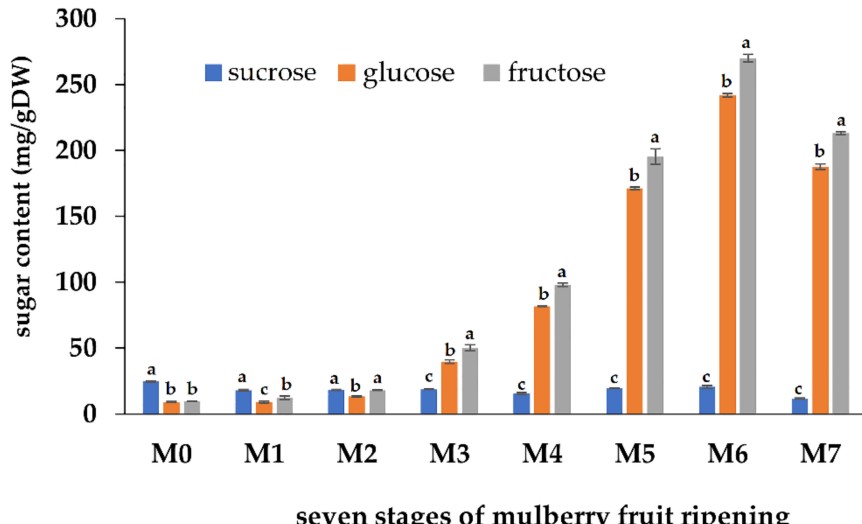

**Figure 2.** Sugar content in mulberry fruit during different ripening stages. Values are expressed as mean ± SD of triplicate measurements (n = 3). M0–M7 are mulberry fruits during ripening stages; means with different letters (a, b, c) are significantly different at $p < 0.05$.

### 3.3. Phenolic Acid and Flavonoid Compounds by HPLC

Individual phenolic acid and flavonoid analyses were performed by HPLC. The results show that the content of total phenolic acids was significantly different ($p < 0.05$) at different ripening stages, as shown in Table 3. The total phenolic acid content increased from M0 to M6, ranging from 903 to 3599 µg/g DW, and decreased slightly at M7 (3358 µg/g DW). Some individual phenolic acids increased, particularly vanillic acid, which was found in mulberry fruit during its ripening stages. The initial amount of vanillic acid was increased by 93% to the highest content in M6. Additionally, protocatechuic acid, caffeic acid, syringic acid and cinnamic acid increased with ripeness, as shown by 6%, 28%, 10% and 6%, respectively. These increased from the lowest content (M0) of individual phenolic acid found in the ripening stages. In contrast, gallic acid, p-hydroxybenzoic acid, chlorogenic acid, p-coumaric acid and sinapic acid decreased with the ripening stages, while ferulic acid showed moderate change. These results are similar to those previously reported; phenolic acids were significantly increased from young to mature stages in ivorian gnangnan (*Solanum indicum* L.) berries [28] and pumpkins (*Cucurbita moschata* Duchesne) [43].

**Table 3.** Changes in phenolic acid contents and flavonoid compounds in mulberry fruit during ripening.

| Parameter | Mulberry Ripening Stage | | | | | | | |
|---|---|---|---|---|---|---|---|---|
| | M0 | M1 | M2 | M3 | M4 | M5 | M6 | M7 |
| *Phenolic acid content (µg/g DW)* | | | | | | | | |
| Gallic acid | 166.37 ± 7.45 [b] | 122.25 ± 0.24 [b] | 112.30 ± 0.15 [b] | 111.55 ± 0.47 [d] | 116.27 ± 1.00 [e] | 113.16 ± 0.71 [f] | 109.75 ± 1.13 [g] | 117.03 ± 0.44 [d] |
| Protocatechuic acid | 22.47 ± 0.06 [i] | 22.98 ± 0.12 [i] | 24.61 ± 0.28 [i] | 30.08 ± 0.42 [i] | 53.36 ± 0.67 [g] | 80.57 ± 2.12 [g] | 134.84 ± 4.59 [d] | 135.03 ± 2.62 [c] |
| *P*-hydroxybenzoic acid | 91.27 ± 0.49 [d] | 105.08 ± 1.50 [c] | 108.13 ± 2.90 [c] | 115.93 ± 2.39 [c] | 149.45 ± 0.54 [d] | 148.13 ± 3.89 [d] | 123.13 ± 0.11 [f] | 99.68 ± 4.66 [e] |
| Chlorogenic acid | 285.39 ± 5.16 [a] | 226.55 ± 3.96 [a] | 213.46 ± 2.59 [a] | 212.45 ± 1.47 [a] | 165.78 ± 2.07 [c] | 143.64 ± 3.36 [e] | 129.06 ± 1.02 [e] | 100.51 ± 30.71 [d] |
| Vanillic acid | 19.38 ± 0.46 [j] | 40.55 ± 2.57 [f] | 69.11 ± 1.94 [f] | 203.23 ± 2.41 [b] | 541.31 ± 0.73 [a] | 1341.45 ± 8.66 [a] | 1769.24 ± 2.16 [a] | 1741.74 ± 5.43 [a] |
| Caffeic acid | 27.47 ± 0.20 [h] | 25.30 ± 1.07 [h] | 39.31 ± 0.74 [h] | 67.56 ± 2.01 [f] | 180.23 ± 6.97 [b] | 410.55 ± 5.74 [b] | 779.34 ± 5.11 [b] | 804.57 ± 12.12 [b] |
| Syringic acid | 9.63 ± 0.20 [k] | 6.46 ± 0.14 [j] | 13.53 ± 0.88 [j] | 19.74 ± 0.31 [j] | 31.77 ± 1.60 [h] | 62.67 ± 2.02 [h] | 103.92 ± 3.01 [h] | 103.18 ± 1.20 [d] |
| *P*-coumaric acid | 81.14 ± 0.28 [e] | 78.07 ± 0.14 [e] | 74.78 ± 0.88 [e] | 61.01 ± 0.31 [h] | 62.27 ± 3.37 [f] | 63.25 ± 5.39 [h] | 67.71 ± 3.94 [i] | 89.15 ± 1.56 [f] |
| Ferulic acid | 63.79 ± 0.32 [f] | 66.25 ± 0.48 [g] | 67.13 ± 0.15 [g] | 65.78 ± 0.18 [f] | 64.60 ± 0.48 [f] | 63.36 ± 0.98 [h] | 64.21 ± 1.30 [j] | 103.87 ± 4.44 [d] |
| Sinapic acid | 101.86 ± 0.75 [c] | 73.31 ± 0.21 [f] | 72.29 ± 0.15 [f] | 62.63 ± 0.15 [g] | 53.19 ± 0.41 [g] | 52.22 ± 3.75 [i] | 22.19 ± 1.39 [k] | 17.72 ± 1.67 [h] |
| Cinnamic acid | 34.23 ± 2.01 [g] | 120.21 ± 9.63 [d] | 101.99 ± 2.39 [d] | 102.25 ± 1.24 [e] | 115.64 ± 4.21 [c] | 211.39 ± 8.11 [c] | 296.01 ± 0.90 [c] | 45.68 ± 0.81 [g] |
| *Total* | 903.02 ± 18.32 [F] | 887.03 ± 22.03 [G] | 896.65 ± 17.16 [G] | 1052.22 ± 13.87 [E] | 1533.87 ± 22.04 [D] | 2690.38 ± 42.73 [C] | 3599.42 ± 24.66 [A] | 3358.15 ± 38.66 [B] |
| *Flavonoid content (µg/g DW)* | | | | | | | | |
| Rutin | 354.88 ± 3.77 [b] | 195.22 ± 9.98 [c] | 110.05 ± 1.94 [e] | 86.78 ± 0.46 [f] | 98.75 ± 0.42 [f] | 93.31 ± 1.71 [e] | 99.34 ± 3.27 [d] | 155.56 ± 2.30 [b] |
| Catechin | 1434.07 ± 22.69 [a] | 1162.55 ± 12.39 [a] | 1012.35 ± 22.31 [a] | 6315.53 ± 81.58 [a] | 7325.23 ± 54.45 [a] | 4145.41 ± 45.99 [a] | 7932.28 ± 38.61 [a] | 8069.68 ± 26.36 [a] |
| Quercetin | 72.59 ± 6.63 [c] | 97.38 ± 5.24 [e] | 117.52 ± 4.68 [d] | 431.04 ± 7.28 [d] | 467.02 ± 10.30 [d] | 104.78 ± 5.16 [d] | 94.13 ± 4.21 [d] | 87.08 ± 2.13 [c] |
| Apigenin | 63.32 ± 1.51 [d] | 137.22 ± 4.29 [d] | 168.19 ± 4.55 [c] | 353.01 ± 4.88 [e] | 391.61 ± 9.76 [e] | 127.37 ± 3.01 [c] | 106.90 ± 4.76 [c] | 72.12 ± 0.65 [d] |
| Myricetin | 47.23 ± 1.25 [e] | 63.72 ± 0.67 [f] | 72.22 ± 1.12 [f] | 582.21 ± 4.18 [c] | 673.69 ± 11.28 [b] | 80.61 ± 4.14 [f] | 46.33 ± 0.44 [e] | 38.44 ± 0.24 [f] |
| Kaempferol | 22.57 ± 0.79 [f] | 1033.56 ± 15.27 [b] | 1063.09 ± 49.92 [b] | 610.17 ± 10.81 [b] | 609.70 ± 8.35 [c] | 369.77 ± 4.93 [b] | 152.77 ± 3.71 [b] | 56.60 ± 1.72 [e] |
| *Total* | 1994.69 ± 36.64 [G] | 2689.66 ± 47.83 [E] | 2543.42 ± 84.51 [F] | 8378.76 ± 109.19 [C] | 9566.01 ± 94.56 [A] | 4921.24 ± 64.93 [D] | 8431.77 ± 55.01 [B] | 8479.49 ± 33.41 [B] |

Values are expressed as mean ± SD of triplicate measurements (n = 3). M0–M7 are mulberry fruit during each ripening stage; means with different letters (a–k) are significantly different at $p < 0.05$ within the same column. Means with different letters (A, B, C, D) are significantly different at $p < 0.05$ within the same row in the parameter.

The flavonoid compounds (rutin, catechin, quercetin, apigenin, myricetin and kaempferol) in mulberry fruit within the different ripening stages are shown in Table 3. The results show that the total flavonoid compounds were found in the range of 1994–9566 µg/g DW for M0 and M4. The predominant flavonoid was catechin and it had the highest content in all ripening stages, while the content of kaempferol was lower than the other samples found in the M0 (22 µg/g DW) ripening stage. Catechin, which belongs to the flavonoid compound group, tended to initially increase to its highest in M4 and then decrease to M7. Similar data were reported by Xin et al. [44] that catechin was the major compound in mulberry fruit of all maturity levels in *M. alba* 'Zhenzhubai' and *M. alba* 'Da10′, and that the active expression of anthocyanidin reductase or leucoanthocyanidin reductase genes in mulberry fruit during maturation caused their contents to increase and decrease [44]. Individual flavonoid compounds found in plants, including mulberry fruit, have been shown to have anti-atherosclerotic, anti-inflammatory, anti-tumor, anti-thrombogenic, anti-osteoporotic, anti-bacterial, anti-viral, anti-fungal, anti-oxidant, anti-platelet, anti-thrombotic action and anti-allergic properties [45,46].

The PCA score plot of phenolic acid content and ripening stages was separated into different blocks and responsible for 54.86% and 44.26% of the total variance of PCA1 and PCA2, respectively (Figure 3a). Furthermore, the ripening stages were variable vectors (seven ripening stages: M0–M7) placed at a slight distance from one another, indicating a strong correlation between them. In addition, M0, M1 and M2 were correlated with the contents of gallic acid, p-hydroxybenzoic acid and chlorogenic acid, while M3, M4, M5, M6 and M7 were correlated with the content of vanillic acid and caffeic acid. This indicates that the content of individual phenolic acids relates to the formation of phenolic acids at different stages, which is similar to a previous study reporting that the first stage of ripening is the initial synthesis of phytochemical compounds and the next stage of ripening changes chemical compounds in mulberry fruits during maturation [47].

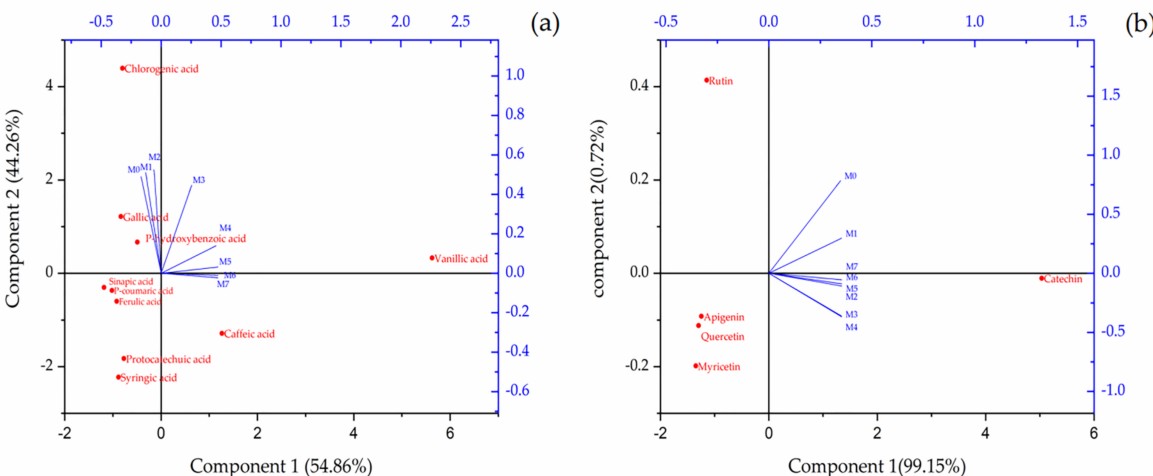

**Figure 3.** (**a**) PCA score plot of phenolic acids and (**b**) PCA score plot of flavonoid compounds. M0 −M7 are mulberry fruits during ripening stages.

The PCA score plot successfully separated the samples for flavonoids content into different blocks, with the first two components (PC1 and PC2) accounting for 99.15% and 0.72% of the total variance, respectively (Figure 3b). Ripening content and flavonoid compounds were investigated. The results show that apigenin, quercetin and myricetin were the predominant compounds found in all samples, as shown on the left half of the plot, while rutin and catechin were separate groups on the right half and upper half, respectively. Additionally, the ripening of M0 to M7 is displayed on the right half of the plot and all mulberry ripening correlated with catechin content as shown by the highest content of catechin in all ripening stages is shown in Table 3. According to the results, the initial ripening was found to have a low concentration of flavonoids due to an increase in flavonoid formation in fruits during maturation [48].

### 3.4. Amino Acid Contents by LC/MS/MS

Amino acids are abundant in some fruits. During the ripening stages of fruit, the amino acid levels increase due to the ripening inducing dramatic changes in the protein and metabolite complex network [49]. Similar to our result, the total amino acids found were in the range 2401–4569 μg/g DW (Table 4). The ripening stages M6 and M5 were shown to have the highest amounts with no significant differences ($p < 0.05$). Meanwhile, M0 had lower than total amino acid content at about half that of M6 and M5. The predominant amino acid in the ripening stages of mulberry fruit was asparagine. In all seven ripening stages, the amount was greater than the other amino acid types. Our study indicates that the amino acid contents in mulberry fruit are influenced by the different ripening stages. When comparing the amount of amino acids with M0, five groups were identified as follows: proline was increased in group 1; arginine, aspartic acid, glutamine, glutamic acid, histidine, lysine, methionine, phenylalanine, serine, threonine and tryptophan increased to the last stage and were moderately decreased in group 2; tyrosine, isoleucine and leucine were slightly changed in group 4; and cysteine was not detected in sample in group 5. The change in the amino acid contents was similar to that previously reported. Boggio et al. [50] and Sorrequieta et al. [51] found that the free amino acid contents in tomato pericarp increased markedly during the ripening stage due to metabolizing enzymes that occurred during the ripening process. However, the amount of chemical compounds, including amino acids, is influenced by harvesting conditions, genotype, growth conditions, genetic diversity and fruit maturity stage [1,31–33]. Therefore, further studies on this aspect are needed for other plants.

The PCA analysis is shown on a score plot of amino acid content and ripening stages. The result was the separation of the display responsible for 88.75% (PCA1) and 11.04% (PCA2) into the different blocks. The results show that Phe, Gln, Lys, Tyr and Asn were abundant in all ripening mulberry fruits and correlate with ripening due to being located at the right half of the block. Furthermore, because of the variable factors displayed at a slight distance from one another, all ripening (M0–M7) was in correlation between them, as shown in Figure 4. These findings explain that the stage of ripening causes the different phytochemicals in each stage, while some individual amino acids, such as Tyr and Asn, can be found in all ripening stages, as shown in Figure 4, and have a high content in all ripening (Table 4).

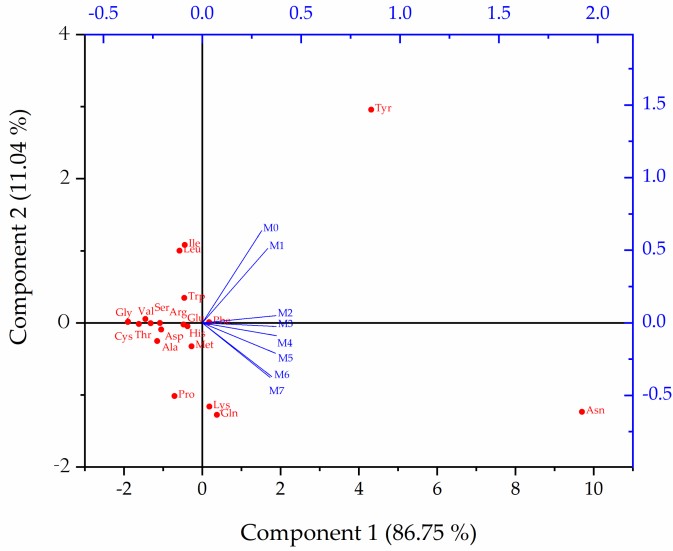

**Figure 4.** PCA score plot of amino acids. M0−M7 are mulberry fruits during ripening stages. Ala: alanine; Arg: arginine; Asn: asparagine; Asp: aspartic acid; Cys: cysteine; Gln: glutamine; Glu: glutamic acid; Gly: glycine; His: histidine; Ile: isoleucine; Leu: leucine; Lys: lysine; Met: methionine; Phe: phenylalanine; Pro: proline; Ser: serine; Thr: threonine; Trp: tryptophan; Tyr: tyrosine; Val: valine.

**Table 4.** Changes in amino acid contents in mulberry fruit during ripening stages.

| Parameter | Amino Acid Content (µg/gDW) | | | | | | | |
|---|---|---|---|---|---|---|---|---|
| | **M0** | **M1** | **M2** | **M3** | **M4** | **M5** | **M6** | **M7** |
| Ala | 27.64 ± 3.79 [k] | 41.01 ± 2.66 [m] | 50.61 ± 3.44 [l] | 51.08 ± 1.37 [l] | 62.29 ± 0.59 [k] | 88.69 ± 2.41 [k] | 131.77 ± 5.49 [i] | 95.70 ± 2.34 [i] |
| Arg | 52.49 ± 2.52 [h] | 68.19 ± 2.08 [i] | 68.94 ± 1.41 [k] | 96.05 ± 2.00 [k] | 85.65 ± 0.66 [i] | 85.83 ± 0.46 [k] | 85.74 ± 1.07 [k] | 77.06 ± 0.68 [j] |
| Asn | 465.55 ± 9.75 [b] | 719.18 ± 11.32 [b] | 1505.70 ± 20.27 [a] | 1690.35 ± 31.41 [a] | 1393.06 ± 30.19 [a] | 1556.81 ± 31.11 [a] | 1193.77 ± 13.58 [a] | 762.77 ± 6.52 [a] |
| Asp | 43.71 ± 1.92 [i] | 56.18 ± 1.50 [j] | 86.59 ± 1.57 [j] | 99.80 ± 5.49 [k] | 85.11 ± 3.99 [i] | 104.03 ± 4.06 [j] | 100.68 ± 0.64 [j] | 78.30 ± 2.55 [j] |
| Cys | ND | ND | ND | ND | ND | ND | ND | ND |
| Gln | 39.33 ± 1.57 [j] | 52.30 ± 1.53 [k] | 96.26 ± 1.39 [i] | 159.48 ± 4.13 [f] | 227.71 ± 6.88 [d] | 403.18 ± 0.41 [c] | 460.47 ± 4.95 [b] | 277.93 ± 8.51 [c] |
| Glu | 96.23 ± 1.80 [f] | 109.90 ± 3.46 [g] | 137.36 ± 3.39 [g] | 136.11 ± 3.67 [h] | 141.34 ± 0.56 [h] | 159.26 ± 3.35 [h] | 164.27 ± 2.93 [h] | 126.73 ± 2.97 [g] |
| Gly | 0.77 ± 0.31 [n] | 0.68 ± 0.11 [p] | 0.37 ± 0.22 [o] | 0.40 ± 0.11 [q] | 0.24 ± 0.18 [o] | 0.49 ± 0.10 [p] | 0.53 ± 0.23 [q] | 0.70 ± 0.14 [p] |
| His | 95.94 ± 8.97 [f] | 113.04 ± 2.20 [g] | 138.81 ± 0.77 [g] | 174.52 ± 25.26 [e] | 164.82 ± 4.95 [f] | 187.20 ± 4.77 [g] | 161.80 ± 2.62 [h] | 130.55 ± 8.66 [g] |
| Ile | 222.46 ± 11.69 [c] | 239.29 ± 3.09 [c] | 184.09 ± 4.10 [e] | 124.27 ± 1.51 [i] | 87.96 ± 1.57 [i] | 87.30 ± 0.55 [k] | 71.64 ± 1.05 [m] | 53.54 ± 0.99 [l] |
| Leu | 207.33 ± 3.05 [d] | 216.76 ± 3.09 [d] | 162.69 ± 3.64 [f] | 113.03 ± 2.12 [j] | 79.94 ± 0.98 [j] | 78.49 ± 0.48 [l] | 64.14 ± 1.80 [n] | 48.33 ± 0.91 [m] |
| Lys | 36.96 ± 1.64 [j] | 48.71 ± 1.07 [l] | 88.16 ± 0.44 [j] | 149.29 ± 2.90 [g] | 205.57 ± 3.02 [e] | 367.61 ± 1.02 [d] | 421.29 ± 4.48 [c] | 254.32 ± 5.80 [e] |
| Met | 74.20 ± 6.32 [g] | 99.42 ± 4.49 [h] | 129.95 ± 3.12 [h] | 157.57 ± 3.64 [f] | 153.60 ± 7.36 [f] | 252.11 ± 11.32 [f] | 229.81 ± 6.89 [f] | 149.70 ± 6.48 [f] |
| Phe | 102.16 ± 6.33 [e] | 167.41 ± 3.39 [e] | 246.67 ± 7.86 [c] | 290.19 ± 2.00 [e] | 258.95 ± 7.20 [c] | 277.00 ± 7.19 [e] | 208.61 ± 1.42 [g] | 109.08 ± 3.01 [h] |
| Pro | 12.83 ± 0.98 [m] | 18.08 ± 0.28 [o] | 19.06 ± 0.40 [n] | 20.29 ± 0.71 [p] | 27.51 ± 0.11 [n] | 83.04 ± 0.81 [k] | 262.78 ± 1.53 [e] | 317.77 ± 1.14 [b] |
| Ser | 46.04 ± 2.31 [i] | 49.32 ± 0.73 [k] | 43.43 ± 0.43 [m] | 36.08 ± 2.26 [n] | 39.65 ± 1.31 [l] | 58.60 ± 0.59 [m] | 80.87 ± 1.39 [l] | 65.14 ± 2.90 [k] |
| Thr | 16.86 ± 1.12 [l] | 19.36 ± 0.20 [n] | 23.63 ± 0.53 [o] | 27.52 ± 0.27 [o] | 26.81 ± 0.58 [n] | 33.61 ± 0.16 [o] | 36.99 ± 4.47 [p] | 25.93 ± 0.31 [o] |
| Trp | 110.10 ± 12.55 [e] | 157.44 ± 0.41 [f] | 197.53 ± 5.86 [d] | 188.89 ± 0.62 [d] | 145.68 ± 1.97 [g] | 145.20 ± 0.97 [i] | 100.42 ± 0.82 [j] | 77.25 ± 1.15 [j] |
| Tyr | 713.36 ± 27.54 [a] | 848.17 ± 13.32 [a] | 688.08 ± 2.46 [b] | 708.47 ± 9.79 [b] | 566.46 ± 12.21 [b] | 555.80 ± 10.86 [b] | 376.06 ± 3.19 [d] | 265.07 ± 1.06 [d] |
| Val | 37.16 ± 1.04 [j] | 41.92 ± 2.93 [m] | 42.74 ± 1.01 [m] | 40.15 ± 1.03 [m] | 36.27 ± 0.34 [m] | 44.73 ± 1.03 [n] | 45.27 ± 1.34 [o] | 42.05 ± 0.15 [n] |
| *total* | 2401.12 ± 103.56 [F] | 3066.36 ± 47.86 [E] | 3910.68 ± 62.31 [C] | 4263.54 ± 100.29 [B] | 3788.62 ± 84.65 [D] | 4568.98 ± 81.55 [A] | 4569.52 ± 59.89 [A] | 2958 ± 56.27 [E] |

Values are expressed as mean ± SD of triplicate measurements (n = 3). ND = not detected; M0–M7 are mulberry fruit during different ripening stages; means with different letters (a–k) are significantly different at *p* < 0.05 within the same column. Ala: alanine; Arg: arginine; Asn: asparagine; Asp: aspartic acid; Cys: cysteine; Gln: glutamine; Glu: glutamic acid; Gly: glycine; His: histidine; Ile: isoleucine; Leu: leucine; Lys: lysine; Met: methionine; Phe: phenylalanine; Pro: proline; Ser: serine; Thr: threonine; Trp: tryptophan; Tyr: tyrosine; Val: valine.

*3.5. Change in γ-Aminobutyric and Anthocyanin in Different Ripening Stages of Mullberry Fruit*

γ-aminobutyric acid, or GABA, is a non-protein amino acid that is found in animals, bacteria and plants [52]. Several previous studies have found GABA in plants like tea, tomato, tobacco and mulberry [52]. In this study, GABA was found in all stages of maturation with a significantly different ($p < 0.05$) range from 94–273 g/gDW in M0 to M7 (Table 5). The trend of GABA content increased when the ripeness level increased. The GABA content of the M7 sample was higher than that of the initial ripening stage (M0) by 12.8 times. Previous studies found that the highest amount of GABA was found in the green stage relative to the amino acid content in tomato [53,54]. However, the content of GABA that occurred was carbolized in the mitochondrial matrix called the GABA shunt, which happened rapidly during the ripening stages [54,55]. Additionally, the GABA level was determined by several factors, such as species, variety, environmental conditions, stress during cultivation and even post-harvest treatments, growth development and ripening stage [52,56]. With respect to GABA, it has been reported as a bioactvie compound with health benefits, including antioxidant, neuroprotective, neurological disorder prevention, anti-hypertensive, anti-diabetic, anti-cancer, anti-inflammatory, antimicrobial, anti-allergic, hepatoprotective and renoprotective [57].

**Table 5.** Changes in γ-aminobutyric acid and anthocyanin content of mulberry fruit during ripening stages.

| Sample Name | γ-Aminobutyric Acid (μg/gDW) | Anthocyanin Content (mg/100 gDW) | |
|---|---|---|---|
| | | Cyaninin-*3*-*O*-glucoside | Peonidin-3 glucoside |
| M0 | 94.53 ± 2.80 [g] | 2.06 ± 0.02 [i] | 0.28 ± 0.01 [h] |
| M1 | 95.56 ± 4.76 [g] | 7.58 ± 0.29 [h] | 0.37 ± 0.01 [g] |
| M2 | 100.75 ± 1.89 [f] | 46.04 ± 0.92 [g] | 0.55 ± 0.03 [f] |
| M3 | 112.72 ± 4.69 [e] | 178.74 ± 0.50 [f] | 1.38 ± 0.07 [e] |
| M4 | 122.97 ± 0.52 [d] | 494.43 ± 2.70 [d,e] | 2.22 ± 0.11 [d] |
| M5 | 177.98 ± 1.50 [c] | 1070.12 ± 7.20 [c] | 3.38 ± 0.18 [c] |
| M6 | 230.27 ± 1.91 [b] | 1902.02 ± 14.44 [a] | 4.61 ± 0.04 [a] |
| M7 | 273.15 ± 1.17 [a] | 1671.22 ± 16.31 [b] | 3.75 ± 0.01 [b] |

Values are expressed as mean ± SD of triplicate measurements (n = 3). M0–M7 are mulberry fruit during different ripening stages; means with different letters (a–g) are significantly different at $p < 0.05$ within the same column.

Anthocyanin is widely found in natural material colorants, as shown by a purple color, and has antioxidant, anti-cancer, anti-diabetic, anti-inflammatory and antibacterial properties [58]. Anthocyanin compounds are useful in food as a natural food colorant, and are also used as purple coloring in cosmetics and medicinal plants. Anthocyanin in mulberry fruit has been reported and the major individual anthocyanin was cyanidin-3-O-glucoside [1]. In this study, it was reported that cyaninin-3-O-glucoside (C3G) was found in all maturity stages of mulberry fruit, while peonidin-3-glucoside (P3G) was also present (Table 5). Additionally, the C3G content was greater than P3G in all samples, confirming previous findings that C3G is abundant in mulberry fruit [59]. The amount of C3G was significantly different ($p < 0.05$), with a range from 2–1902 mg/100 gDW in M0 and M6, and dramatically decreased in M7 (1671 mg/100 gDW). The initial amount of C3G in mulberry fruit rapidly increased during the ripening stage from M0 to M6 due to the anthocyanin biosynthesis pathway, which was possibly induced at a position relatively far downstream from the late stage reactions catalyzed by enzymes, including dihydroflavonol 4-reductase (DFR), anthocyanidin synthase (ANS) or anthocyanidin 3-O-glucosyltransferase in the latter stages of development (UFGT) [9]. C3G is obvious in fruit as a red hue. In mulberry fruit, M2 had a red color on the fruit skin, which indicates that the C3G content was moderately increased and then decreased (M7), similar to the peonidin-3-O glucoside content.

### 3.6. Antioxidant Activity

The results of the antioxidant activity with mulberry fruit during all seven ripening stages are shown in Figure 5. The objective of the antioxidant study was to evaluate the amount of DPPH radicals that were scavenged using the Trolox equivalents on centration (TE) and the FRAP valuation (mg FeSO$_4$/g DW). The chemical DPPH is a free radical. The color of the DPPH has been used for a long time to evaluate the capacity to scavenge free radicals. When antioxidants interact with the DPPH radical, which is detected using a microplate reader, they cause a color shift from purple to yellow, measured by absorbance at 517 nm. The result shows that the radical scavenging activity of DPPH was found to be highest in the last ripening stage (M7) at 19.8 mgTE/gDW and lowest in M3 (7.98 mg TE/gDW) (Figure 5a). These results indicate that the values of DPPH radical-scavenging activity of mulberry fruit during the seven ripening stages are increased compared with the initial stage (M0) due to the synthesis of some phytochemicals, especially anthocyanin, when leveled up. As reported by previous authors, the high capacity of anthocyanin can scavenge the radical of DPPH [1,58]. The green ripening stage (M0-M3) had higher vitamin C content compared with other ripening stages. These compounds, also found in other fruit, perform as natural antioxidants. Moreover, these reports of anti-oxidant activity were assessed using the FRAP method to determine how effectively they reduced molecules that function by donating an atom from hydrogen to break the chain of the free radicals [60], as shown in Figure 5b. The results were similar to the DPPH activity. The highest amount of FRAP was in the last ripening stage M7 (22 mg FeSO$_4$/gDW) and the lowest was in M3 (7 mg FeSO$_4$/gDW). The antioxidant with DPPH and FRAP assay displayed a similar pattern to the total phenolic content (TPC) and total flavonoid content (TFC) in the mulberry fruit ripening stage. This result indicates that the TPC and TFC are positively correlated with antioxidants with the DPPH and FRAP assays (Table 6), previously supported by this report [15]. Nevertheless, the relationship between DPPH/FRAP and TPC/TFC is not always compatible due to some of the phenolic acids and flavonoid compounds having stronger reducing power than others [4].

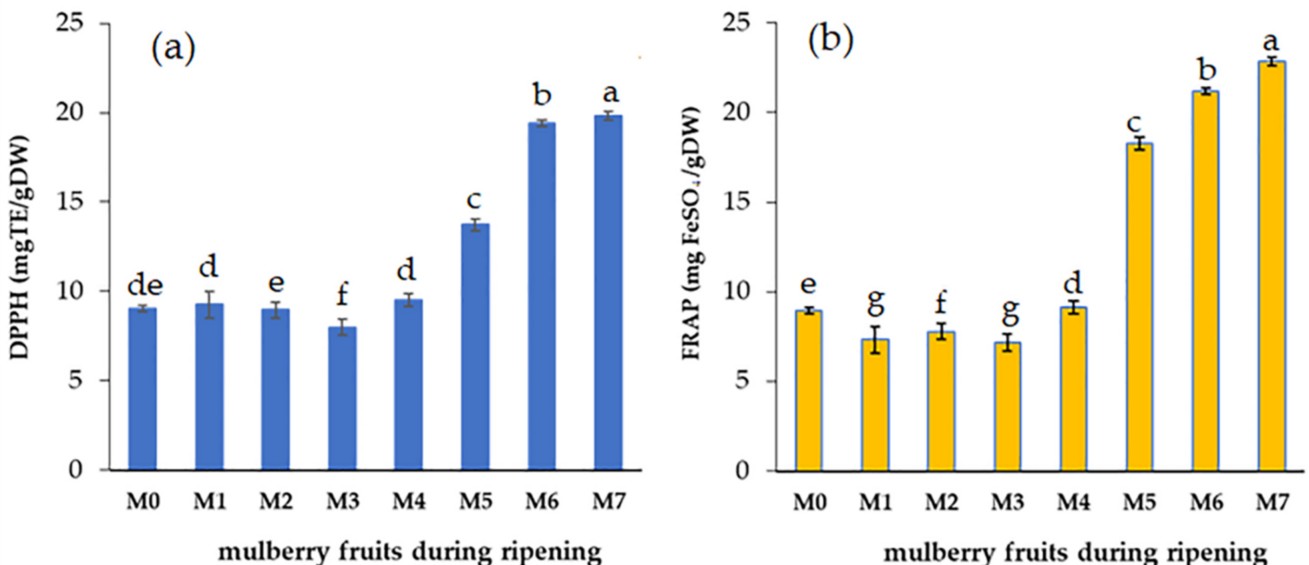

**Figure 5.** Changes in antioxidant activities measured by means of DPPH radical scavenging (**a**) and FRAP analyses (**b**) of mulberry fruit during different ripening stages. M0–M7 are mulberry fruit during different ripening stages; means with different letters (a–g) are significantly different at $p < 0.05$ within the same column.

**Table 6.** Correlations between total phenolic content, total flavonoid content, DPPH scavenging and FRAP activities of mulberry fruits with ripening.

|  | TPC | TFC | DPPH | FRAP |
|---|---|---|---|---|
| TPC | 1 | 0.810 ** | 0.974 ** | 0.936 * |
| TFC | - | 1 | 0.792 ** | 0.813 * |
| DPPH | - | - | 1 | 0.970 ** |
| FRAP | - | - | - | 1 |

TPC: Total phenolic content; TFC: Total flavonoid content; DPPH radical scavenging activities FRAP: Ferric reducing antioxidant activities. **. Correlation is significant at the 0.01 level (2-tailed); *. Correlation is significant at the 0.05 level (2-tailed).

## 4. Conclusions

In conclusion, mulberry fruit in the seven different ripening stages differ significantly ($p < 0.05$) in phytochemical compounds (including phenolic acids, flavonoids, anthocyanins, organic acids and sugars) and antioxidant activity (DPPH and FRAP). In this regard, each ripening stage is a potential source of phytochemical compounds, such as TPC, TFC and sugar content, with the highest content in M6, total phenolic acid content highest in M5, total flavonoid compounds highest in M4, total amino acid contents highest in M6, GABA content highest in M7, anthocyanin content highest in M6 and antioxidant content highest in M7. At different ripening stages, mulberry fruit may be suited for goods based on the desired ingredients, such as vitamin C, amino acids, anthocyanin, phenolic acid, flavonoid chemicals or antioxidant capacity. The results of this study should be helpful in designing products that utilize mulberry fruit while maintaining its high antioxidant content, excellent bioactivity and quality for use in food, cosmetics and medicine.

**Supplementary Materials:** The following supporting information can be downloaded at: https://www.mdpi.com/article/10.3390/horticulturae8121146/s1, Table S1: Multiple Reaction Monitoring (MRM) conditions for amino acids and anthocyanin on LC/MS/MS., Table S2: The importance of the studies on phytochemical compounds in mulberry fruits.

**Author Contributions:** T.C. and S.S. methodology; T.C., C.P. and R.S. formal analysis; T.C., C.P. and R.S. data curation; T.C. and S.S. writing—original draft preparation; T.C. and S.S. writing—review and editing; T.C. and S.S. funding acquisition. All authors have read and agreed to the published version of the manuscript.

**Funding:** This research was funded by Mahasarakham University.

**Institutional Review Board Statement:** Not applicable.

**Informed Consent Statement:** Not applicable.

**Data Availability Statement:** All data generated or analyzed are contained within the present article.

**Acknowledgments:** The authors thank the Laboratory Equipment Center of Mahasarakham University for cooperation and scientific assistance and the Silk Innovation Center at Mahasarakham University for providing mulberry fruits as raw materials. Also, thanks to Jolyon Dodgson, who is an agriculturist, crop scientist and plant pathologist from the UK, for English proofreading.

**Conflicts of Interest:** The authors declare no conflict of interest.

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
