# Peer review of "Evaluation of Color, Phytochemical Compounds and Antioxidant Activities of Mulberry Fruit (Morus alba L.) during Ripening"

_horticulturae, doi:10.3390/horticulturae8121146_

Round 1

Reviewer 1 Report

Do not use acronyms in abstract, without detailed description.

It should be specifically mentioned, in 2.1 Color determination, if the samples were dried or not during this measurement, with more details about the temperature gradient which can change the colour properties.

Also the colour differece, ΔE, should be calculated and the sample values  compared between themselves and with the colour significant difference threshold of ΔEth = 5 units.

Figure 2: should change the sucose in sucrose.

All the results have to be represented in graphs (like Figure 2), and some comparisons with relative differences can facilitate the analysis.

Correlations between the parameters are missing.

These are results from a tedious work, so chemometrics should/have to be involved in the analysis, in order to complete the research. Also, due to time series measurements, linear/nonlinear regressions can be applied and variable time-variation (rate) can be established.

Author Response

Responses to reviewers

Manuscript number: horticulturae-2007476

Title: Evaluation of color, phytochemical compounds, and antioxidant activities of   mulberry fruit (Morus alba L.) during ripening

Dear Editor and reviewers,

We have taken all the comments into account and revised throughout the manuscript (please see the attachment). A list of changes and corrections are as below;

Reviewer 1

We would like to thank the reviewer for their kind assessment of our manuscript. All comments will be taken into account and responded thoroughly the manuscript.

Q1: Do not use acronyms in abstract, without detailed description.

Ans: Thank you for your suggestion. We have done as suggested by reviewer. (Blue mark)

Q2: It should be specifically mentioned, in 2.1 Color determination, if the samples were dried or not during this measurement, with more details about the temperature gradient which can change the colour properties.

Ans:  We have done as suggested by reviewer. We are measuring of color in fresh samples and we added the sentence in section 2.1 Plant materials and sample preparation (line 69-78) and 2.1 Color determination (line 84) (Blue mark)

Q3: Also the colour differece, ΔE, should be calculated and the sample values compared between themselves and with the colour significant difference threshold of ΔEth = 5 units.

Ans: We have done as suggested by reviewer. Please see on Table 1. We added method of calculated ΔE (line 91-92) and it was added in text on discussion line 231-235. (Blue mark)

 We added reference no.27 in discussion. Please see on reference no.27.

  1. Lou, H.; Hu, Y.; Zhang, L.; Sun, P.; Lu, H. Nondestructive Evaluation of the Changes of Total Flavonoid, Total Phenols, ABTS and DPPH Radical Scavenging Activities, and Sugars during Mulberry (Morus Alba L.) Fruits Development by Chlorophyll Fluorescence and RGB Intensity Values. LWT - Food Science and Technology 2012, 47, 19–24, doi:10.1016/j.lwt.2012.01.008.

Q4:Figure 2: should change the sucose in sucrose.

Ans:  We have done as suggested by reviewer. Please see on Figure 2.

Q5: All the results have to be represented in graphs (like Figure 2), and some comparisons with relative differences can facilitate the analysis.

Ans: Thank you for suggestion. We have done as suggested by reviewer on table 5 to graph (Figure 3). Please see on Figure 3.  However, some data on Tables 1-4 were very different data, it cannot present in graph, but it was shown in Table form better.

Q6: Correlations between the parameters are missing.

Ans:  We sorry for mistake without table of correlations. So, we added the table 5 of correlations between total phenolic content, total flavonoid content, DPPH scavenging activities and FRAP assay of mulberry fruits with ripening. Please see on Table 5 and line 416.

Q7: These are results from a tedious work, so chemometrics should/have to be involved in the analysis, in order to complete the research. Also, due to time series measurements, linear/nonlinear regressions can be applied and variable time-variation (rate) can be established.

Ans: Thank you for suggestion. We would like to thank the reviewer for their kind assessment of our manuscript. We found many data, i.e. color, phytochemical compounds (phenolic acids, flavonoids, anthocyanin, amino acids, γ-aminobutyric and antioxidant activities of mulberry fruit during ripening. This data should be helpful in designing products that utilize mulberry fruit while maintaining its high antioxidant content, excellent bioactivity and quality for use in food, cosmetics and medicine

In addition to all comments, we have revised the typo errors throughout the manuscript.

Thank you for your kind consideration,

Sincerely yours,

Chumroenphat, T

Reviewer 2 Report

The manuscript entitled: “Evaluation of color, phytochemical compounds, and antioxidant activities of mulberry fruit (Morus alba L.) during ripening”, addresses a topic related to bioactive compounds of mulberry.

This article is of clear interest, but must be improved significantly with some of the suggested modifications I have included below.

The following aspects should be addressed: Extensive editing of English language and style is required. There are parts of the article difficult to understand due to the writing is not adequate. Abstract Line 24: Please change “GABA” by γ-aminobutyric, this is important because a reader that only will read the abstract will not know the meaning of GABA.

Line 27: The abstract indicated that the antioxidants were found to have the highest contents in M7 which were positively correlated with the total phenolic and total flavonoid contents., however in lines 23 and 24 is indicated that the higher content of flavonoids was found in M6. How is it possible this correlation if the higher content of flavonoids was found in M6? Please, explain it in the abstract and in the results.

Introduction

In several parts of the manuscript is used a comma before “and”, for example in the line 37: (leaf, stem, root, and fruit), the comma before “and” is incorrect please remove.

Lines 41 and 42: The sentence indicate that “phytochemical compounds of mulberry fruit have been reported from many countries”. The countries are not the ones reporting it, this information is report by studies performed by researchers, please improve the sentence.

Line 45: Please change “samples” by “this fruit”.

Line 51 and 52: This sentence not have sense: “There have been several reports of mulberry fruit with certain colors (green, red, purple, white, and black)[12].” What do the authors mean?

Line 59: Please check the sentence. This sentence not have sense. What is the meaning of “the frequency on this sentence?

Materials and Methods

Please it is necessary add some important information to this epigraph:

What software was used for do the statistical analysis?

What method was used for do the correlations?

From which part of the tree were the samples picked up?

On what date was each sample picked up?

Line 72: What is the meaning of “were identified by Dr. Surapon Saensouk? Do you mean that those trees were not known to be mulberry trees? Please remove this sentence, this information is irrelevant.

Line 95: Please change “test” by “content” Line 120: This sentence is not adequately written, please correct it. Line 124: What is the meaning od “DI”, please indicate it. Line 159: Please change “was performed following Lee…” by “was performed following the method described by Lee…”

Results and Discussion

Please it is necessary include a table with information about most important climatic parameter (maximum temperature, minimum temperature, humidity…) during the time that was performed the study.

Line 206, 207 and 208: This information: “In general, the phytochemical compounds studied were phenolic acid, flavonoid compounds, vitamin C, free sugar, amino acids, and anthocyanin along with antioxidant activity”, was included in Materials and Methods, this information is repetitive, remove it.

Line 244: I do not understand why the results are compared with the result obtained with blueberry. A good discussion must compare the results with other studies of mulberry, in the case that compare with other species, must be compare with several species instead one specie. Line 256: Why the results are compared with guava and amla? Line 307: What is the meaning of “Da10” in this sentence? Table 3: Indicate the meaning of “ND”.

Line 407 and 408: The sentence indicated that "result indicates that the TPC and TFC were positively correlated with antioxidants with the DPPH and FRAP assays", please show a table with this correlation.

Author Response

Responses to reviewers

Manuscript number: horticulturae-2007476

Title: Evaluation of color, phytochemical compounds, and antioxidant activities of  mulberry fruit (Morus alba L.) during ripening

Dear Editor and reviewers,

We have taken all the comments into account and revised throughout the manuscript (please see the attachment). A list of changes and corrections are as below;

Reviewer 2

This article is of clear interest, but must be improved significantly with some of the suggested modifications I have included below.

We would like to thank the reviewer for their kind assessment of our manuscript. All comments will be taken into account and responded thoroughly the manuscript.

The following aspects should be addressed: Extensive editing of English language and style is required. There are parts of the article difficult to understand due to the writing is not adequate. 

Q1: Abstract Line 24: Please change “GABA” by γ-aminobutyric, this is important because a reader that only will read the abstract will not know the meaning of GABA. 

Ans:   Thank you for suggestion. We have done as suggested by reviewer. Please see on line 23. (Blue mark)

Q2: Line 27: The abstract indicated that the antioxidants were found to have the highest contents in M7 which were positively correlated with the total phenolic and total flavonoid contents., however in lines 23 and 24 is indicated that the higher content of flavonoids was found in M6. How is it possible this correlation if the higher content of flavonoids was found in M6? Please, explain it in the abstract and in the results.

Ans:   Sorry, we are apologies, For TPC, TFC and γ-aminobutyric increased when the ripening level increased. Wile, amino acids, anthocyanin and vitamin C increased initially, peaking in M6 and then declined in the final ripening stage (M7). We are change in abstract. Please see on line 23. (Blue mark)

Introduction

Q3: In several parts of the manuscript is used a comma before “and”, for example in the line 37: (leaf, stem, root, and fruit), the comma before “and” is incorrect please remove.

Ans: Thank you for your suggestion. We have deleted as suggested by reviewer.

Q4: Lines 41 and 42: The sentence indicate that “phytochemical compounds of mulberry fruit have been reported from many countries”. The countries are not the ones reporting it, this information is report by studies performed by researchers, please improve the sentence.

Ans: Thank you for your suggestion. We changed countries to researcher and added references “[1–5]” in text. Please see on line 42. (Blue mark)

Q5: Line 45: Please change “samples” by “this fruit”.

Ans: Thank you for your suggestion. We have changed as suggested by reviewer. Please see on line 45.(Blue mark) 

Q6: Line 51 and 52: This sentence not have sense: “There have been several reports of mulberry fruit with certain colors (green, red, purple, white, and black)[12].” What do the authors mean?

Ans: It means the different color of mulberry fruits. Therefore, this sentence has been changed to “There have been several reports of mulberry fruit with different colors (green, red, purple, white and black colors)[12]” (lines 51-52) (Blue mark)

Q7: Line 59: Please check the sentence. This sentence not have sense. What is the meaning of “the frequency on this sentence?

Ans: “The frequency” on this sentence has been deleted.

Materials and Methods

Please it is necessary add some important information to this epigraph:

Q8: What software was used for do the statistical analysis?

Ans: We used SPSS software for statistical analysis and we added this sentence in section 2.10. Statistical analysis. “All statistical analyses were performed using SPSS for Windows” Please see on line 207-208. (Blue mark)

Q9: What method was used for do the correlations?

Ans: We used SPSS software and Pearson’s correlation test. Please see on line 207. (Blue mark)

“The pearson’s correlation test to assess the correlations among the means.”

Q10: From which part of the tree were the samples picked up?

Ans: The part of this study used mulberry fruits with seven stage ripening and we improved the section 2.1. Please see on line 69-78. (Blue mark)

From

“The seven stages of mulberry fruit ripening were identify based on fruit color and previous studies[13] (Figure 1). The mulberry cultivar was Chaingmai-60 and the fruit were harvested in January-February, 2022, from the Silk Innovation Center Mahasarakham University, Thailand. The plants were identified by Dr. Surapon Saensouk, plant taxonomist from the Walai Rukhavej Botanical Research Institute, Mahasarakham University, Mahasarakham, Thailand. The specimens were deposited in the herbarium with the voucher specimen number of CT190201-08. Fresh fruit were washed before removing the seeds from the fruit pulp. For each ripening stage, 500 g of fruit was dried using a freeze dryer (Scanvac CoolSafe model 100-9 Pro, LaboGene ApS, Denmark) until the moisture content of the samples were less than 7%. The ground samples were sieved through a 40-mesh wire sieve and kept at -20 OC until analyzed.”

To

 “The part of mulberry fruit used in this study was the Chaingmai-60 cultivar, grown on a farm by the Silk Innovation Center at Mahasarakham University, Thailand. The hand-harvested at seven stages of mulberry fruit ripening were identify based on fruit color and previous studies [13] on 21 January 2022 (Figure 1). The specimens were deposited in the herbarium with the voucher specimen number of CT190201-08. For each ripening stage, 500 g of fresh fruit were measured for skin color. Then, they were washed and dried using a freeze dryer (Scanvac CoolSafe model 100-9 Pro, LaboGene ApS, Denmark) until the moisture content of the samples was less than 7%. The ground samples were sieved through a 40-mesh wire sieve and kept at -20 OC until analyzed for phytochemical compounds and antioxidant activities.”

Q11: On what date was each sample picked up?

Ans: We picked up on 21 January 2022. Please see on line 72. (Blue mark)

Q12: Line 72: What is the meaning of “were identified by Dr. Surapon Saensouk? Do you mean that those trees were not known to be mulberry trees? Please remove this sentence, this information is irrelevant.

Ans: We have deleted as suggested by reviewer. Please see on line 69-78. (Blue mark)

Q13: Line 95: Please change “test” by “content” 

Ans: We have changed as suggested by reviewer. Please see on line 95. (Blue mark)

Q14: Line 120: This sentence is not adequately written, please correct it.

Ans: We have improved as suggested by reviewer. We added the sentence. Please see on line 211- 214. (Blue mark)

“The extraction of sugar by mixing 0.1 g of powdered dry samples with 5 mL of distilled water and incubating the mixture in a water bath at 80°C for 30 min. For HPLC analysis, the supernatants from the samples were collected and filtered through a 0.22 mm filter after being centrifuged at 6300 x g for 10 min.”

 Q15: Line 124: What is the meaning od “DI”, please indicate it. 

Ans: Thank you for your suggestion. We have changed “DI” to “Deionized water”. Please see on line 127. (Blue mark)

Q15: Line 159: Please change “was performed following Lee…” by “was performed following the method described by Lee…”

Ans: Done.

Results and Discussion

Q16: Please it is necessary include a table with information about most important climatic parameter (maximum temperature, minimum temperature, humidity…) during the time that was performed the study.

Ans: Thank you for your suggestion. We added information of temperature and relative humidity at Mahasarakham university at 21 January 2022 in section 2.1 2.1 Plant materials and sample preparation.  Please see on line 72-73. (Blue mark)

Q17: Line 206, 207 and 208: This information: “In general, the phytochemical compounds studied were phenolic acid, flavonoid compounds, vitamin C, free sugar, amino acids, and anthocyanin along with antioxidant activity”, was included in Materials and Methods, this information is repetitive, remove it.

Ans: Done.

Q18: Line 244: I do not understand why the results are compared with the result obtained with blueberry. A good discussion must compare the results with other studies of mulberry, in the case that compare with other species, must be compare with several species instead one specie.

Ans: Thank you for your suggestion. We deleted TFC comparison with blueberry and added the compared with mulberry fruit in the other countries. Pleased see on line 252-253. (Blue mark)

 Change the reference from

Huang, W.; Zhang, H.; Liu, W.; Li, C. Survey of Antioxidant Capacity and Phenolic Composition of Blueberry, Blackberry and Strawberry in Nanjing. J. Zhejiang Univ. Sci. B 2012, 13, 94–102, doi:10.1631/jzus.B1100137.

To

Kamiloglu, S.; Serali, O.; Unal, N.; Capanoglu, E. Antioxidant Activity and Polyphenol Composition of Black Mulberry (Morus Nigra L.) Products. Journal of Berry Research 2013, 3, 41–51, doi:10.3233/JBR-130045.

Please see on reference no.29

 Q19: Line 256: Why the results are compared with guava and amla?

Ans: This sentence “These contents were lower than 1.8 times that of guava (22 mg/g)[37] and 5.5 times that of fresh amla fruit (66 mg/g)[38], which the source of vitamin c from plant” was deleted from text. And we added the sentence in text.

“The previously reported vitamin C content of Chinese mulberry fruit was 2.5 mg/g DW, and that of Egyptian black mulberry was 1.3 mg/g FW. These contents were lower than those found in the present study, at 4.9 and 9.4 times, respectively.” (line 263-266). (Blue mark)

We added reference order 36-37 ( Please see on reference)

  1. Wang, K.; Kang, S.; Li, F.; Wang, X.; Xiao, Y.; Wang, J.; Xu, H. Relationship between Fruit Density and Physicochemical Properties and Bioactive Composition of Mulberry at Harvest. Journal of Food Composition and Analysis 2022, 106, 104322, doi:10.1016/j.jfca.2021.104322.
  2. Aly, A.A.; Ali, H.G.M.; Eliwa, N.E.R. Phytochemical Screening, Anthocyanins and Antimicrobial Activities in Some Berries Fruits. Food Measure 2019, 13, 911–920, doi:10.1007/s11694-018-0005-0.

Q20: Line 307: What is the meaning of “Da10” in this sentence? 

Ans: “Da10” was variety of Mulberry (M. alba ‘Da10’) in Xin et al. studied. It has been changed in text. Please see on line315. (Blue mark)

Reference:  Xin, Y.; Meng, S.; Ma, B.; He, W.; He, N. Mulberry Genes MnANR and MnLAR Confer Transgenic Plants with Resistance to Botrytis Cinerea. Plant Science 2020, 296, 110473, doi:10.1016/j.plantsci.2020.110473.

Q21: Table 3: Indicate the meaning of “ND”.

Ans:  Thank you for your suggestion. We added information of ND to footnote. “ND = not  detected”.  Please see on Table 3. (Blue mark)

Q22: Line 407 and 408: The sentence indicated that "result indicates that the TPC and TFC were positively correlated with antioxidants with the DPPH and FRAP assays", please show a table with this correlation.

Ans: Sorry. We are mistake for table of correlations. We added the table of correlations between total phenolic content, total flavonoid content, DPPH scavenging activities and FRAP assay of mulberry fruits with ripening. Please see on Table 5 and line 416. (Blue mark)

In addition to all comments, we have revised the typo errors throughout the manuscript.

Thank you for your kind consideration,

Sincerely yours,

Chumroenphat, T

Round 2

Reviewer 2 Report

The manuscript have been improved following the reviewer sugestions, however climatic parameter (maximum temperature, minimum temperature, humidity) during the time was performed the study (from fruit set to harvest day) are not been included. Please include a table with this information.

Author Response

Dear reviewers,

We have taken all the comments into account and revised throughout the manuscript (please see the attachment). 

Thank you for your kind consideration,

Sincerely yours,

Chumroenphat, T
